# Erucin, an H_2_S-Releasing Isothiocyanate, Exerts Anticancer Effects in Human Triple-Negative Breast Cancer Cells Triggering Autophagy-Dependent Apoptotic Cell Death

**DOI:** 10.3390/ijms24076764

**Published:** 2023-04-05

**Authors:** Ivana Bello, Martina Smimmo, Roberta d’Emmanuele di Villa Bianca, Mariarosaria Bucci, Giuseppe Cirino, Elisabetta Panza, Vincenzo Brancaleone

**Affiliations:** 1Department of Pharmacy, School of Medicine and Surgery, University of Naples Federico II, 80131 Naples, Italy; 2Department of Science, University of Basilicata, 85100 Potenza, Italy

**Keywords:** hydrogen sulfide, triple-negative breast cancer, erucin, apoptosis, autophagy, MDA-MB-231

## Abstract

Breast cancer is the most frequent form of cancer occurring in women of any age. Among the different types, the triple-negative breast cancer (TNBC) subtype is recognized as the most severe form, being associated with the highest mortality rate. Currently, there are no effective treatments for TNBC. For this reason, the research of novel therapeutics is urgently needed. Natural products and their analogs have historically made a major contribution to pharmacotherapy and the treatment of various human diseases, including cancer. In this study, we explored the potential anti-cancer effects of erucin, the most abundant H_2_S-releasing isothiocyanate present in arugula (Eruca sativa) in MDA-MB-231 cells, a validated in vitro model of TNBC. We found that erucin, in a concentration-dependent manner, significantly inhibited MDA-MB-231 cell proliferation by inducing apoptosis and autophagy. Additionally, erucin prevented intracellular ROS generation promoting the expression of key antioxidant genes and halted MDA-MB-231 cell migration, invasion, and colony formation. In conclusion, using a cellular and molecular biology approach, we show that the consumption of erucin could represent a novel and promising strategy for intervention against TNBC.

## 1. Introduction

Triple-negative breast cancer (TNBC) accounts for about 12–20% of all breast cancer cases and affects with higher frequency young women carrying BRCA1 gene expression [1]. The term triple-negative refers to the fact that breast cancer cells lose the expression of three major classes of hormone receptors: estrogen receptor (ER), progesterone receptor (PR), and human epidermal growth factor receptor 2 (HER2). These alterations make patients refractory to conventional hormonal therapies, causing major difficulties in clinical decision making and management. Current treatment options are limited to surgery, adjuvant chemotherapy, and radiotherapy. Although progress has been made over the last century in cancer research, TNBC still causes the highest rate of deaths among all types of breast cancer [2]. For this reason, there is an urgent need to discover and develop new therapeutics to treat effectively TNBC. In this regard, numerous therapeutic dietary or non-dietary compounds present in plants have gained much interest due to their preventive and protective action against different types of cancer [3,4,5]. A considerable number of pre-clinical and clinical studies show that phytochemicals exert protective effects against the growth and spread of tumor cells by regulating their proliferative capacity, and survival, as well as counteracting the production of reactive oxygen species (ROS) [6,7]. Among edible phytochemicals, the consumption of cruciferous vegetables belonging to the large family of Brassicaceae (also called Cruciferae) has been reported to decrease the risk of developing several types of cancers, including lung, colorectal, ovarian, pancreatic, and breast cancer [8,9,10]. The Brassicaceae family includes the rocket plant, which is an annual plant very popular all around the world for the edible green leaves commonly used for salad and food dish recipes. The two main varieties of rocket plants are Diplotaxis tenuifolia (wild rocket) and Eruca sativa (arugula, or cultivated rocket). Eruca sativa contains a wide range of compounds with nutraceutical and organoleptic properties, including isothiocyanates, phenolic compounds, and carotenoids. Among them, erucin (4-methylthiobutyl isothiocyanate) is one of the major components of leaves obtained from the enzymatic hydrolysis of glucoerucin. As demonstrated in a large number of in vitro and in vivo studies, erucin exerts protective effects against several types of human cancers, including breast cancer [8,11,12,13,14,15]. Remarkably, recent findings revealed that the protective effect of erucin relies on its intrinsic capacity to slowly release hydrogen sulfide (H_2_S), which is one of three gasotransmitters, along with nitric oxide (NO) and carbon monoxide (CO), which act in our body to modulate numerous biological processes [16]. 

Despite that, the potential use of erucin against TNBC has not yet been investigated. Toward this goal, in this study we explored the potential anticancer effect of erucin on cell proliferation and survival, ROS production, migration, and invasion in human MDA-MB-231 cells using an in vitro model of TNBC.

## 2. Results

### 2.1. Erucin Inhibits Proliferation and Induces Apoptosis in Human TNBC Cells

To investigate the potential effect of erucin in human TNBC cells, we measured the proliferation capacity of MDA-MB-231 cells exposed to a crescent concentration of erucin using the MTT assay. As shown in Figure 1A, erucin in a concentration-dependent manner (1–100 µM) inhibited the growth of MDA-MB-231 cells after 48 h with an IC50 of about 24 µM. Subsequently, we explored the potential effects of erucin on cell necrosis and apoptosis in the same cell line using FACS analysis. Using this approach, we found that erucin (30 μM) induced both early and late cell apoptosis in about 60% of MDA-MB-231 cells (Figure 1B). This effect was then confirmed by western blot analysis, revealing the time-dependent cleavage (hence activation) of Caspase-3 and its substrate poly (adenosine diphosphateribose) polymerase (PARP) by erucin (Figure 1D).

### 2.2. Erucin Promotes Autophagy in Human TNBC Cells

In the next stage, we explored whether erucin could modulate autophagy in MDA-MB-231 cell; this is an evolutionarily conserved cellular process that is importantly intertwined with apoptosis and plays a critical role in regulating cancer-cell development and proliferation [17]. We found that erucin (30 μM), again in a time-dependent manner, promoted autophagy in MDA-MB-231 cells; this was reflected by the increased expression of key regulatory genes, including ULK1, ATG13, BECN1, and BNIP3 (Figure 2A–D). Additionally, by western blot analysis, we found that erucin significantly increased the expression of LC3II protein, which is the lipidated form of the LC3I (Figure 2F,G), and concomitantly decreased the expression of p62. These latter are known to be two key positive and negative regulators of autophagosome formation and activity, respectively (Figure 2F–H).

### 2.3. Erucin Reduces ROS Production by Inducing the Expression of Antioxidant Genes in Human TNBC Cells

Cumulative evidence shows that the unbalanced formation of reactive oxygen (ROS) species is one of the major pathological mechanisms underpinning cancer development and progression [18]. In light of this, a key step in this study was to understand if the protective effect of erucin observed in MDA-MB-231 cells could also be ascribed to the formation of ROS species. Toward this goal, MDA-MB-231 cells were exposed to H_2_O_2_/Fe^2+^ (2 mM), which is an efficient method to increase the production of ROS in in vitro cell model systems. We found that pre-treatment for 3 and 6 h with erucin (30 μM) in MDA-MB-231 cells exposed to H_2_O_2_/Fe^2+^ significantly prevented ROS formation as determined by the DCF fluorescence intensity (Figure 3A,B). To investigate the mechanism through which erucin was capable of preventing the ROS formation in TNBC cells, we measured the expression of key genes encoding for antioxidant enzymes, including heme oxygenase-1 (HMOX-1), Glutamate-Cysteine Ligase Catalytic Subunit (GCLC), Glutamate-Cysteine Ligase Modifier Subunit (GCLM), and Superoxide Dismutase 1 (SOD1). Treatment with erucin 30 μM for 3 and 6 h significantly increased the expression of all the aforementioned genes in a time-dependent manner (Figure 3C,D). In addition, we found that erucin induced the translocation from the cytoplasm to the nucleus of NF-E2–related factor 2 (Nrf2), which is one of the best-known transcription factors regulating the expression of numerous antioxidant genes; it is also concomitantly reduces the expression of its repressor Kelch-like ECH-associated protein 1 (Keap1) (Figure 3E,F).

### 2.4. Erucin Inhibits Cell Motility, Invasion, and Colony Formation of Human TNBC Cells

In the expriment’s final stage, we evaluated the potential antimetastatic activity of erucin in MDA-MB-231 cells performing wound healing and cell invasion assays. Erucin 0.3 and 1 μM concentrations (selected for not interfering with cell proliferation and apoptosis) prevented migration and invasiveness of MDA-MB-231 cells compared to control vehicle-treated cells (Figure 4A,B and Figure 5A,B). Additionally, erucin significantly decreased colony formation ability, reducing both their number and diameter (Figure 4A,B and Figure 5A,B).

## 3. Discussion

Natural compounds, including those present in Cruciferae, are shown to significantly prevent and decrease cancer development and progression. In particular, the protective effect of Cruciferae is attributed to isothiocyanates, a large class of products derived by the hydrolysis of glucosinolates, including sulforaphane and erucin [19]. Remarkably, in the last few years, the potential use of erucin as a new strategy to tackle chemoresistance in cancer emerged owing to its ability to remove oxidative stress and halt inflammatory signaling [20,21,22]. In a recent review article, Singh and colleagues report that erucin exerts anticancer effects by inhibiting carcinogen-activating enzymes and promoting the activity of detoxifying enzymes and the mechanisms leading to the inhibition of metastasic processes and angiogenesis [23]. Additionally, erucin leads to the activation of apoptotic processes in cancer cells by inducing cell cycle arrest and suppressing microtubule dynamics [24,25]. However, erucin potentiates the anticancer activity of 4-hydroxytamoxifen in estrogen receptor-positive breast cancer cell lines [26] and, in combination with lapatinib, seems to overcome drug resistance in the same cells [27,28]. 

H_2_S donors and their derivatives were demonstrated to effectively reduce the proliferation and apoptosis of TNBC cells by inhibiting the phosphorylation or expression of proteins associated with NF-κB, PI3K/Akt/mTOR and Ras/Raf/MEK/ERK signaling pathways. They also inhibited the expression of MMP-2/9 and EMT to resist TNBC metastasis by inhibiting aberrant activation of the β-catenin pathway [29]. Recently, Citi et al. reported that erucin limits the proliferation, growth, and survival of pancreatic cancer cells through a mechanism dependent on the slow release of H_2_S [13]. However, the role of H_2_S in cancer is still matter of debate since, in the current relevant literature, there are studies either supporting a beneficial or detrimental role on cancer development, differentiation, metastasis, and proliferation. These divergent effects are most likely due to different effect in different type of cancers as well as the methodological approaches used. Another variant to be taken into account is the release kinetic of H_2_S. Indeed, H_2_S is known to produce hormetic responses, which refer to a biphasic dose/concentration effect. Typically, a low dose produces stimulatory or beneficial effects, while a high dose produces inhibitory or toxic effects [30,31,32]. The second important issue is how much of the hydrogen sulfide released effectively penetrates the cells. Erucin it has been shown to release H_2_S inside cells in a dose-dependent manner [33]. In our experimental settings, erucin reduced proliferation in a concentration-dependent manner of MDA-MB-231 cells, which is an elective model of TNBC with an high proliferative capacity. In particular, at the concentration selected 30 µM erucin causes an inhibition of cell proliferation of about 50%. Moreover, at this concentration erucin also promotes a significant apoptosis concomitantly enhancing the expression of key genes and proteins (BECN1, LC3, ULK1, ATG13, BNIP3) triggering and driving autophagy. Thus, erucin acts on both autophagy and apoptosis, with these functions being intimately intertwined to control, among numerous processes, the cell fate [34,35]; this approach represents a promising target in many types of cancer. Intracellular imbalance in ROS production is one of the major pathological mechanisms underpinning tumor initiation and metastatic progression [36]. Treatment of cells with erucin significantly prevented the intracellular formation of ROS in TNBC cells. This effect could be simply due to the ability of the H_2_S released intracellularly by erucin to scavenge ROS directly [37]. However, based on our results, the antioxidant effect of erucin is not only exerted through the direct “quenching” effect of H_2_S on ROS; erucin also induces the expression of Nrf2 and concomitantly reduces the expression of its repressor Keap1. Nrf2 is also a key transcriptional factor acting to promote the expression of numerous antioxidant genes, including GCLC, GCLM, SOD1, and HMOX-1. Since the expression of all the aforementioned genes was increased in MDA-MB-231 cells, erucin displays a direct quenching effect coupled to a molecular modulation on Nrf2, as shown by other researchers [38]. To further characterize the erucin profile, we studied the ability of this H_2_S donor to interfere in vitro with migration, invasion, and colony formation of MDA-MB-231 cells. To perform this study, we used doses that do not trigger apoptosis and/or autophagy. Erucin, at the 1 µM concentration, showed a significant inhibitory effect on migration. The same dose caused an even more marked inhibitory effect on invasion and colony formation, giving about 80% on both parameters evaluated. These results further support the proposal that erucin effects involve a reduction in both ROS and more than one molecular mechanism.

In conclusion, our study provides novel insights into the use of hydrogen sulfide donors as a promising strategy for curative and/or adjuvant therapies against TNBC. Moreover, our data suggest that low and slowly delivered levels of hydrogen sulfide intracellularly are necessary to achieve beneficial effect. However, future studies are necessary to explore the effect of erucin alone or in combination with conventional anticancer agents using in vivo models of TNBC. 

## 4. Materials and Methods

### 4.1. Cell Culture and Reagents

Human triple-negative breast cancer cell line MDA-MB-231 (cat. no. HTB-26) was obtained from the American Type Culture Collection (ATCC, Manassas, VA, USA). Cells were cultured in DMEM (cat. D6546, Sigma-Aldrich, Milan, Italy) complemented with 10% fetal bovine serum (FBS) (Gibco, Milan, Italy; cat. no. A4736301), penicillin (100 U/mL), streptomycin (100 μg/mL) (cat. no. 30-002-CI), 2 mmol/l L-glutamine (cat. no. 25-005-CI), and 0.01 M HEPES (cat. no. 25-060-CI) (all from Corning, Manassas, VA, USA). Cells were maintained in a humified incubator at 37 °C with 5% CO_2_. Erucin (cat. 4430-36-8) was obtained from Cayman Chemical (MI, USA). 

### 4.2. Western Blot Analysis

Total proteins were obtained from cells using a lysis RIPA buffer (50 mM Tris-HCl (pH 7.4), 1 mM EDTA, 150 mM NaCl, 1% Triton X-100, and 0.25% (*v*/*v*) sodium deoxycholate, 0.1% SDS) supplemented with Protease Inhibitor Cocktail (cat. P8340-1 mL, Merk, Milan, Italy). Protein lysates were quantified by the Bradford method (cat. 5000006, Bio-Rad, Italy). An equal amount of proteins (40 μg) from each cell extract was loaded and separated on sodium dodecyl sulphate polyacrylamide (SDS-PAGE) gel; it was then transferred on a nitrocellulose membrane using Trans-Blot Turbo Transfer Starter System (Bio-Rad, Italy). After transfer, the membranes were blocked in 5% low-fat milk in PBS with 0.1% Tween 20 (PBST) for 1 h at room temperature before being incubated overnight at 4 °C with the following primary antibodies: (i) Caspase 3 (cat. 9662, Cell Signaling, MA, USA; diluted 1:1000); (ii) PARP (cat. 9542, Cell Signaling, MA, USA; diluted 1:1000); (iii) LC3 (cat. 2775, Cell Signaling, MA, USA; diluted 1:1000); (iv) p62 (cat. 5115, Cell Signaling, MA, USA; diluted 1:1000); (v) Nrf2 (cat. sc-722; Santa Cruz Biotechnology, Santa Cruz, CA; diluted 1:500); (vi) α-tubulin (cat. 3873; Cell Signaling, MA, USA; diluted 1:1000); (vii) Keap1 (cat. 8047; Cell Signaling, MA, USA; diluted 1:1000); and (viii) β-actin (cat. sc-47,778; Santa Cruz Biotechnology, Santa Cruz, CA; diluted 1:1000). β-actin and α-tubulin were used as the control normalizing protein for cytosolic and nuclear lysates, respectively. The membranes were washed three times with PBST and incubated with anti-mouse (cat. 115-035-003) or anti-rabbit (cat. 111-035-144) IgG secondary antibodies (Jackson ImmunoResearch, Cambridge, UK, dilution 1:3000) for 1.30 h at room temperature. The membranes were then washed again three times with PBST. Protein bands were visualized using the ECL chemiluminescence method (Clarity TM Western ECL Substrate, cat. 1705061, Bio-Rad Laboratories, CA, USA) and Chemidoc XRS (Biorad, Milan, Italy). Quantification of results was determined using ImageJ Software (version 1.52a, U.S. National Institutes of Health).

### 4.3. RNA Purification and Quantitative Real-Time PCR

Total RNA was extracted from cells using the QIAzol Lysis Reagent according to the manufacturer’s instructions (cat. 79306, Qiagen, Hilden, Germany). Spectrophotometric quantization of each purified RNA was estimated using a Nanodrop apparatus (Thermo Fisher Scientific, MA, USA). The purified RNA was considered DNA and protein-free if the ratio between readings at 260/280 nm was ≥1.8. Isolated mRNA was reverse-transcribed using iScript Reverse Transcription Supermix for RT-qPCR (cat. 1708841, Bio-Rad, Italy). qPCR was carried out in a CFX96 real-time PCR detection system (Bio-Rad) with the use of SYBR Green master mix kit (cat. 1725271, Bio-Rad) and the following primers:

ULK1

5′-AGCACGATTTGGAGGTCGC-3′

5′-GCCACGATGTTTTCATGTTTCA-3′

ATG13

5′-TTGCTATAACTAGGGTGACACCA-3′

5′-CCCAACACGAACTGTCTGGA-3′

BECN1

5′-ACCTCAGCCGAAGACTGAAG-3′

5′-AACAGCGTTTGTAGTTCTGACA-3′

BNIP3

5′-ATGTCGTCCCACCTAGTCGAG-3′

5′-TGAGGATGGTACGTGTTCCAG-3′

HMOX-1

5′-GCCGTGTAGATATGGTACAAGGA-3′ 

5′-AAGCCGAGAATGCTGAGTTCA-3′ 

GCLC

5′-GTTGGGGTTTGTCCTCTCCC-3′

5′-GGGGTGACGAGGTGGAGTA-3′

GCLM

5′-AGGAGCTTCGGGACTGTATCC-3′

5′-GGGACATGGTGCATTCCAAAA-3′

SOD1

5′-GGTGGGCCAAAGGATGAAGAG-3′

5′-CCACAAGCCAAACGACTTCC-3′

The real-time PCR cycling protocol was: (i) polymerase activation and DNA denaturation 95 °C for 30 s; (ii) amplification for 40 cycles; (iii) denaturation for 15 s at 95 °C; (iv) annealing and extension for 30 s 60 °C; (v) plate read at 60 °C; and (vi) melt-curve analysis performed at 65–95 °C in 0.5 °C increments at 5 s/step. The housekeeping gene ribosomal protein S16 (5′-TACCACGGAGGCCACCTAA-3′; 5′-CTGCTCCACAAATCGGCCAT-3′) was used as an internal control to normalize the CT values, using the 2^−ΔΔCt^ formula. 

### 4.4. MTT Proliferation Assay

Cell proliferation was measured by MTT (3-(4,5-dimethylthiazol-2-yl) 2,5-diphenyltetrazolium bromide) assay. MDA-MB-231 cells were seeded on 96-well plates (6 × 10^3^ cells/well). After 24 h, cells were treated with erucin 1, 3, 10, 30 and 100 µM for 48 h. The medium was then replaced with 100 µL/well of DMEM containing 0.25 mg/mL of MTT (cat. M5655, Merk, Italy). The plate was incubated for 3 h at 37 °C. The medium was then removed from the wells and 100 µL/well of DMSO (cat. 20-139, Sigma-Aldrich, Milan, Italy) was added and stirred for 15 min; the absorbance at 490 nm was measured using a microplate spectrophotometer (Thermo Scientific Multiskan GO, Thermo Fisher Scientific, MA, USA).

### 4.5. Flow Cytometry Analysis

Apoptosis was measured using the BD Pharmingen™ FITC Annexin V Apoptosis Detection Kit I (cat. 556547, BD Biosciences, Franklin Lakes, NJ, USA) according to the manufacturer’s instructions. MDA-MB-231 cells were seeded (2.5 × 10^5^ cells/well) in 35 mm culture dishes. The day next, the cells were either treated with erucin (30 μM) for 48 h or left untreated. MDA-MB-231 cells were later collected and stained with Annexin V-FITC/Propidium Iodide (PI). Flow cytometry analysis was performed using a BriCyte flow cytometer (Mindray, Italy). A minimum of 50,000 events for each sample were collected and data were analyzed using FlowJo v10 software (Tree Star, Ashland, OR USA).

### 4.6. Wound Healing Assay

MDA-MB-231 cells were seeded in 6-well plates (3 × 10^5^ cells/well). Once the cells reached 90% of confluence, erucin 0.3 and 1 µM were added, and a wound area was created by scraping the cell monolayer with a sterile 200 μL pipette tip. Subsequently, the cells were incubated at 37 °C in 5% CO_2_. The width of the wounded area was photographed with an inverted microscope (20-fold magnification) at the time zero point as well as after 24 h and 48 h of treatment. The wounded area was measured using Image J software (version 1.52a, U.S. National Institutes of Health).

### 4.7. Clonogenic Assay

MDA-MB-231 cells were seeded in 6-well plates (1 × 10^3^ cells/well) on the next day, treated with erucin (0.3 and 1 μM), and allowed to form colonies for 14 days. After washing with PBS, MDA-MB-231 colonies were fixed with 4% paraformaldehyde and colored with 0.5% crystal violet. Colonies composed of more than 50 cells were manually counted under the microscope and captured by a digital camera.

### 4.8. Invasion Assay

Boyden chambers with polycarbonate filters and a nominal pore size of 8 μm (cat. PIEP12R48, Millipore, USA) were coated on the upper side with Matrigel (Becton Dickinson Labware, USA). The chambers were placed in a 24-well plate and MDA-MB-231 cells (2.5 × 10^5^ cells/mL) were plated in the upper chamber in the presence or absence of erucin (0.3 and 1 μM) in serum-free DMEM. At the end of the 16 h incubation period, the medium was removed, the filters were fixed with 4% formaldehyde for 2 min, and the cells were permeabilized with 100% methanol for 20 min. The methanol was removed and the chambers were stained with Giemsa for 15 min; they were then washed with PBS. The filters were removed and the nonmigrating cells on the top of the filter were removed using a cotton swab. The filters were then placed on a slide and examined under a microscope. Cell invasion was determined by counting the number of cells stained on each filter in at least 4–5 randomly selected fields.

### 4.9. Intracellular ROS Measurement

The generation of intracellular reactive oxygen species (ROS) was estimated using the fluorescence probe 2′,7′-Dichlorfluorescein (DCF, cat. 35848, Sigma-Aldrich, Milan, Italy). For the experiments, MDA-MB-231 cells were plated in 96-multi-well black plates (Corning, USA) at a density of 10^4^ cells/well and, after reaching 80% of confluence, were incubated with erucin 30 μM for 3 and 6 h. After washing with 100 µL/well of PBS, cells were incubated for 30 min with 200 μL of 10 μM DCF in HBSS containing 1% FBS. Finally, cells were washed and incubated with Fenton’s reagent (H_2_O_2_/Fe^2+^ 2 mM) for 3 h at 37 °C. The DCF fluorescence intensity was detected using a fluorescent microplate reader (excitation 485 nm and emission 538 nm; GloMax^®^-Multi Detection System, Promega). The intracellular ROS levels were expressed as relative fluorescence intensity after normalizing to viable cell numbers. Additionally, the fluorescence images of differently treated cells were observed by fluorescence microscopy after washing twice with PBS. DCF was observed to emit green fluorescence.

### 4.10. Statistical Analysis

Data were expressed as mean ± SEM of *n* = 3 experiments. Data were analyzed and presented using GraphPad Prism 8.2.0 software (San Diego, CA, USA). Significance was determined using Student’s 2-tailed t-test. Results were considered significant at *p* values less than 0.05 and are labelled with a single asterisk. In addition, *p*-values lower than 0.01 and 0.001 are indicated with double and triple asterisks, respectively.

## Figures and Tables

**Figure 1 ijms-24-06764-f001:**
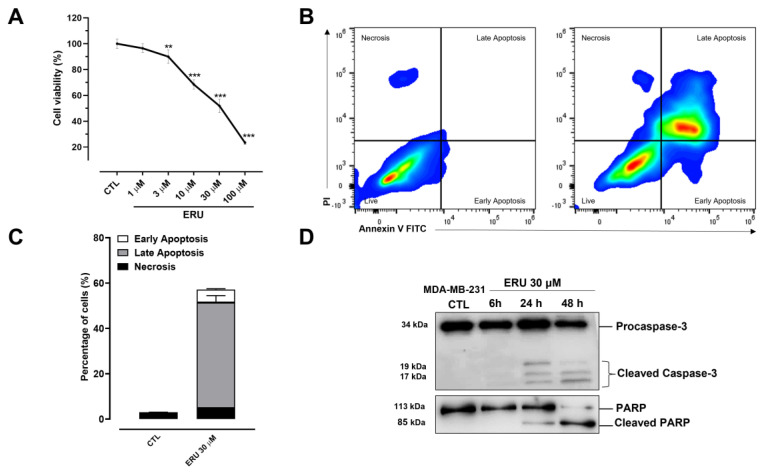
ERU induces apoptosis in MDA-MB-231 cells. (**A**) MTT proliferation assay was performed in MDA-MB-231 cells treated with ERU 1−100 μM for 48 h. (**B**) Flow cytometric analysis of apoptosis detected by Annexin V/propidium iodide (PI) staining in MDA-MB-231 cells treated with ERU (30 μM) for 48 h. (**C**) Quantitative analysis of apoptosis showing that at 48 h, about 60% of MDA-MB-231 cells treated with ERU exhibit markers of apoptosis. (**D**) Western blot analysis of Caspase-3 and PARP on MDA-MB-231 whole-cell lysates after treatment with ERU (30 μM) for 6, 24, and 48 h. The data shown are representative of three independent experiments (*n* = 3) with similar results. ** *p* < 0.01; *** *p* < 0.001 vs. CTL.

**Figure 2 ijms-24-06764-f002:**
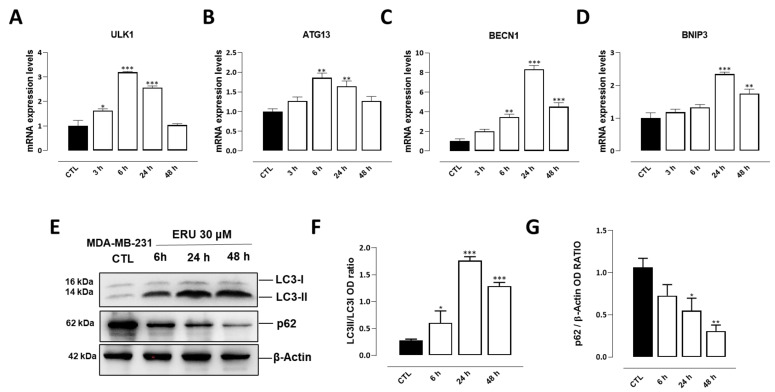
ERU induces autophagy in MDA-MB-231 cells. (**A**–**D**) mRNA expression levels of ULK1, ATG13, BECN1, and BNIP3 in MDA-MB-231 cells following the treatment with ERU (30 μM) for 3, 6, 24, and 48 h; (**E**) representative blots and quantitative analysis of LC3 (**F**) and p62 (**G**) proteins measured in MDA-MB-231 following the treatment with ERU (30 μM) for 6, 24, and 48 h. β-Actin was detected as a loading control. Each data point was obtained from three independent determinations for each experimental condition. * *p* < 0.05; ** *p* < 0.01; *** *p* < 0.001 vs. CTL.

**Figure 3 ijms-24-06764-f003:**
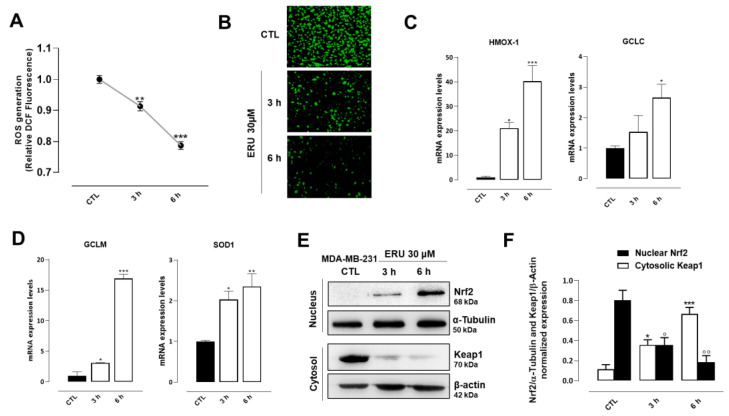
ERU reduces ROS production and promotes the expression of antioxidant enzymes in MDA-MB-231 cells throughout the Nrf2 pathway. (**A**) Intracellular ROS levels in MDA-MB-231 cells treated with Fenton’s reagent (2 mM H_2_O_2_/Fe^2+^) alone or in presence of ERU (30 μM) for 3 and 6 h. Values are expressed as relative DCF fluorescence intensity normalized to viable cell numbers. (**B**) Representative fluorescence microscopic images of intracellular ROS production by DCF staining (green) in MDA-MB-231 cells. (**C**,**D**) mRNA expression levels of HMOX-1, GCLC, GCLM, and SOD1 in MDA-MB-231 cells treated with ERU (30 μM) for 3 and 6 h. (**E**,**F**) Representative blots and relative quantitative analysis of nuclear Nrf2 and cytosolic Keap1 protein expression in MDA-MB-231 cells treated with ERU (30 μM) for 3 and 6 h. α-tubulin and β-actin were detected as a loading control for nucleic and cytosolic portions, respectively. Results are shown as mean ± SEM of three independent experiments (*n* = 3). * *p* < 0.05, ** *p* < 0.01, *** *p* < 0.001, vs. CTL; ° *p* < 0.05, °° *p* < 0.01, vs. nuclear CTL.

**Figure 4 ijms-24-06764-f004:**
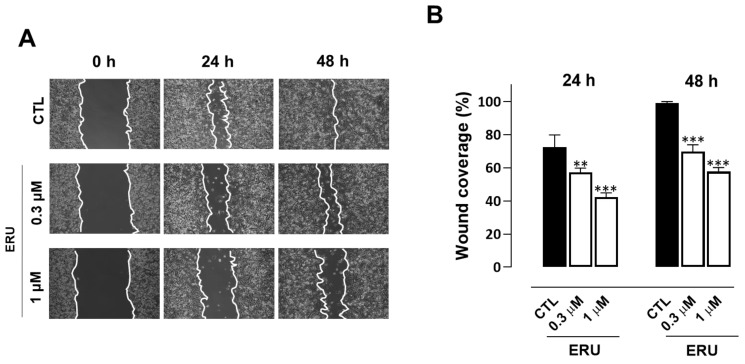
ERU inhibits the migration of MDA-MB-231 cells. (**A**) Representative photographs of migratory MDA-MB-231 cells treated with ERU (0.3 and 1 μM) at 0, 24 and 48 h. (**B**) The scratched areas were quantified in three random fields in each treatment. Data are shown as mean ± SEM of three independent experiments (*n* = 3). ** *p* < 0.01; *** *p* < 0.001 vs. CTL.

**Figure 5 ijms-24-06764-f005:**
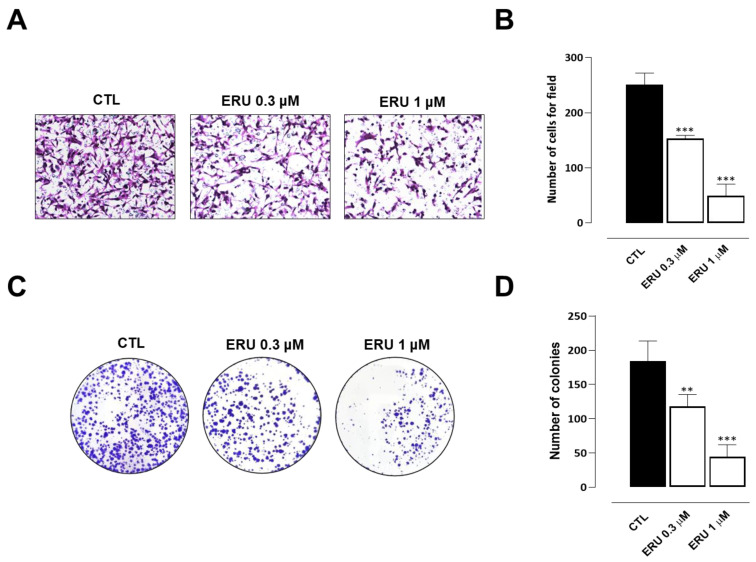
The effect of ERU on MDA-MB-231 cell invasion and colony formation. (**A**) Representative photographs and average number (**B**) of invasive MDA-MB-231 cells treated with ERU (0.3 and 1 μM). (**C**) Representative photographs and average number (**D**) of MDA-MB-231 cells colony formation with ERU (0.3 and 1 μM). Data are shown as mean ± SEM of three independent experiments (*n* = 3). ** *p* < 0.01; *** *p* < 0.001 vs. CTL.

## Data Availability

The data are contained within the manuscript.

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
