# Peer review of "Erucin, an H2S-Releasing Isothiocyanate, Exerts Anticancer Effects in Human Triple-Negative Breast Cancer Cells Triggering Autophagy-Dependent Apoptotic Cell Death"

_ijms, 2023, doi:10.3390/ijms24076764_

Round 1

Reviewer 1 Report

Breast cancer is the most frequent form of cancer occuring in women of any age. Due to this fact, the research of novel therapeutics is urgently needed. In the present study, the authors investigated the anti-cancer effects of erucin, the most abundant H2S-releasing isothiocyanate present in Eruca sativa in MDA-MB-231 cells, a valiated in vitro model of triple-negative breast cancer (TNBC). Data show that erucin significantly inhibited MDA-MB-231 cell proliferation by inducing apoptosis and autophagy. Additionally, erucin prevented intracellular ROS generation promoting the expression of key anti-oxidant genes like expression of transcription factor Nrf2 and downregulation of Keap2.

Finally, the authors showed that erucin also halted MDA-MB-231 cell proliferation, invasion, and colony formation.

However, future studies are necessary to explore the effects of erucin using in vivo models of TNBC.

Author Response

We are grateful to the reviewer for appreciating our study and fully agree that future studies especially in animal models of TNBC are necessary to confirm the beneficial effects of erucin.  

Reviewer 2 Report

In this manuscript, the authors investigated the anti-cancer effect of erucin in MDA-MB-231 cells, and suggested that it triggers an autophagy-dependent apoptotic cell death. 

However, there are some issues that should be clarified:

1) The authors mentioned "...multidisciplinary approach..." (lines 23, 63)

The methods used as described in Section 4 (lines 213-352) belong to the same field (biochemistry, molecular and cellular biology). However, to be addessed as "multidisciplinary" it should include others such as proteomics, structural biology (NMR, X-ray crystallography, cryoEM), atomic force microscopy, confocal and/or electron microscopy, among others.

2) Results (sections 2.1 and 2.2)

The time points were the authors report the features of apoptosis (section 2.1) and/or autophagy (section 2.2) are clearly different (24 hs vs 6 hs), thus suggesting that it is not a mixed-form of cell death as stated "autophagy-dependent apoptotic cell death" but rather time-dependent sequencial events. Moreover, necrosis is also seen (Fig.1B, 1C) since it is a process which is time- and concentration-dependent. 

How the authors explain this discrepancy.

3) In Discussion section, it will be worth to discuss some earlier work such as Pawlik et al, 2016 which also studied the effects of erucin and reported autophagy and apoptosis in ER-breast cancer cells. 

Pawlik A  et al, Sensitization of estrogen receptor-positive breast cancer cell lines to 4-hydroxytamoxifen by isothiocyanates present in cruciferous plants. Eur J Nutr. 2016 Apr;55(3):1165-80. 

4) In the Discussion section, some important references linking erucin and apoptosis in various cellular models including breast cancer are missing. It will be worth to discussed them accordingly.

Below are listed some of them:

- G. Li et al, Mitochondrial translocation and interaction of cofilin and Drp1 are required for erucin-induced mitochondrial fission and apoptosis, Oncotarget 6 (2015) 1834–1849.

- Prełowska M et al, 4-(Methylthio)butyl isothiocyanate inhibits the proliferation of breast cancer cells with different receptor status. Pharmacol Rep. 2017 Oct;69(5):1059-1066.

- Kaczyńska A, et al, Sensitization of HER2 Positive Breast Cancer Cells to Lapatinib Using Plants-Derived Isothiocyanates. Nutr Cancer. 2015;67(6):976-86. 

- Kaczyńska A, Herman-Antosiewicz A. Combination of lapatinib with isothiocyanates overcomes drug resistance and inhibits migration of HER2 positive breast cancer cells. Breast Cancer. 2017 Mar;24(2):271-280.

- Herz C et al, The isothiocyanate erucin abrogates telomerase in hepatocellular carcinoma cells in vitro and in an orthotopic xenograft tumour model of HCC. J Cell Mol Med. 2014 Dec;18(12):2393-403. 

- Azarenko O et al, Erucin, the major isothiocyanate in arugula (Eruca sativa), inhibits proliferation of MCF7 tumor cells by suppressing microtubule dynamics. PLoS One. 2014 Jun 20;9(6):e100599. 

- Lamy E, Mersch-Sundermann V. MTBITC mediates cell cycle arrest and apoptosis induction in human HepG2 cells despite its rapid degradation kinetics in the in vitro model. Environ Mol Mutagen. 2009 Apr;50(3):190-200. 

- Jakubikova J et al, Isothiocyanates induce cell cycle arrest, apoptosis and mitochondrial potential depolarization in HL-60 and multidrug-resistant cell lines. Anticancer Res. 2005 Sep-Oct;25(5):3375-86. 

- Doudican NA et al, Enhancement of arsenic trioxide cytotoxicity by dietary isothiocyanates in human leukemic cells via a reactive oxygen species-dependent mechanism. Leuk Res. 2010 Feb;34(2):229-34.

5)  In the Discussion section, it will be worth to compare the results obtained in this study and earlier work also in MDA-MB-231 cells and other breast cancer cell lines. Particularly, a recent paper by Singh et al. 2020 summarizes the molecular targets of erucin and clearly shows the events seen at different time points.  

Singh D, et al. Molecular targets in cancer prevention by 4-(methylthio)butyl isothiocyanate - A comprehensive review. Life Sci. 2020 Jan 15;241:117061. 

Author Response

We thank the reviewer for his/her constructive comments and suggestions. Below is a point-by-point response to each request raised. 

(1) We apologize for not being more accurate. Following the suggestion, we changed the sentence by deleting "using a multidisciplinary approach". (page 1, line 23; page 2, line 62). 

(2) We would draw the reviewer's attention to the fact that no discrepancies are reported in our study.  As shown in Fig 1 and 2 both autophagy and apoptosis in MDA-MB-231 cells were measured at the same time points (6-24 and 48h). The antioxidant effect of ERU was instead evaluated at only 3 and 6 h before the apoptosis induction. 

Regarding the low amount of cells in necrosis observed in FACS analysis, we believe that it was an effect dependent on ERU. However, it was a negligible effect compared to that of apoptosis in MDA-MB-231 cells at least at the concentration and time used in our experimental conditions. 

(3-4-5) According to the request, we added a few sentences (highlighted in yellow) in the discussion to cite most of the studies indicated (lines 162-169; pag.6). We thank the reviewer for the precious suggestion. 

Round 2

Reviewer 2 Report

The authors clearly addressed all comments raised by the reviewer; thus improving the quality of the manuscript